# Mechanism and Emergence of Stacked Attention Heads in Multi-Layer Transformers

**Tiberiu Mușat**
ETH Zürich, Switzerland
Giotto.ai, Switzerland
`tmusat@ethz.ch`

## Abstract

In this paper, I introduce the *retrieval* problem, a simple yet common reasoning task that can be solved only by transformers with a minimum number of layers, which grows logarithmically with the input size. I empirically show that large language models can solve the task under different prompting formulations without any fine-tuning. To understand how transformers solve the retrieval problem, I train several transformers on a minimal formulation. Successful learning occurs only under the presence of an implicit curriculum. I uncover the learned mechanisms by studying the attention maps in the trained transformers. I also study the training process, uncovering that attention heads always emerge in a specific sequence guided by the implicit curriculum.

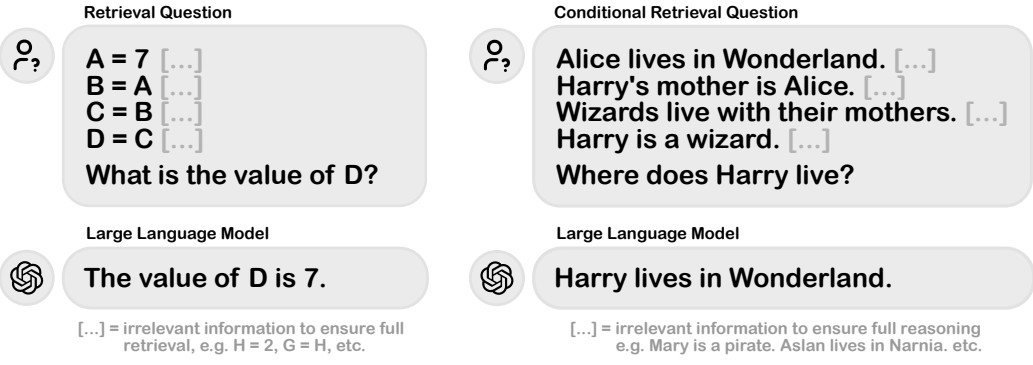

Figure 1: Illustrative examples of *retrieval* and *conditional retrieval* questions.

## 1 Introduction

How do neural networks solve the tasks that they are trained on? Is there a clear algorithm hiding behind the millions of unintelligible weights and biases? These are the questions that the field of mechanistic interpretability tries to answer. If successful, this line of research could lead to a better understanding of neural networks and the development of AI systems with increased safety, reliability, and efficiency (Doshi-Velez & Kim, 2017; Olah et al., 2020; Elhage et al., 2021).

Transformers (Vaswani, 2017) have become the dominant architecture in natural language processing, achieving state-of-the-art results on a wide range of tasks (Brown, 2020; Achiam et al., 2023). Recent interpretability research has successfully uncovered the mechanisms learned by single-layer (Nanda et al., 2023; Quirke et al., 2023) and two-layer (Olsson et al., 2022) transformers. However, understanding the mechanisms learned by deeper transformers remains an open problem. Automatic circuit analysis of large language models provides valuable insights about isolated circuits that span several layers, but such circuits remain not fully understood (Wang et al., 2022; Conmy et al., 2023). Therefore, understanding the mechanisms of multi-layer transformers is a crucial step towards understanding state-of-the-art language models.

## 2 MY CONTRIBUTION

In this paper, I try to answer the following questions:

**Q1.** Are there tasks that can be solved only by transformers with a specific depth?
**Q2.** Are large language models able to solve such tasks without specific fine-tuning?
**Q3.** What is the mechanism that transformers use to solve the task?
**Q4.** How does this mechanism emerge during training?

I answer **Q1** positively by introducing the *retrieval* problem, as well as a close variant that I term the *conditional retrieval* problem (Section 3). I answer **Q2** positively in Section 4 by empirically showing that large language models can solve both problems without any specific fine-tuning under multiple prompting formulations. In Section 5, I provide a formal proof that the retrieval problem requires a minimum number of transformer layers that is logarithmic in the input size.

This suggests that large language models have learned a complex mechanism formed by multiple stacked attention heads. To elucidate this mechanism (**Q3**), I train several transformers on a minimal formulation of the retrieval problem (Sections 6 and 7). In Section 8, by studying the attention maps in the trained transformers, I uncover multiple possible mechanisms that I term *retrieval heads*. Regarding the training process (**Q4**), I find that retrieval heads emerge only under the presence of an implicit curriculum and always in a specific sequence (Section 9).

The retrieval problem also has important implications for **emergent abilities** in large language models (Wei et al., 2022), which I discuss in Section 10.

## 3 THE RETRIEVAL PROBLEM

### 3.1 DEFINITION

The *retrieval* problem is directly inspired by the *induction* problem introduced by Olsson et al. (2022). Given a sequence of tokens $\ldots ab \ldots a$, the induction problem requires the model to predict the token $b$. I directly extend this formulation by increasing the number of induction steps to $D$. Given an input sequence $\ldots x_{D-1} x_D \ldots \ldots \ldots x_1 x_2 \ldots x_0 x_1 \ldots x_0$, the retrieval problem consists in predicting the token $x_D$. By setting $D = 1$, we recover exactly the original induction problem. Throughout this paper, I also refer to the tokens in the retrieval chain using capital letters (i.e., **A** for $x_0$, **B** for $x_1$, **C** for $x_2$, and so on).

I also propose a more general variant of the retrieval problem, which I term the *conditional retrieval* problem, where each retrieval step could depend on multiple previously retrieved values, not just the last one. For example, given the input sequence $\ldots xyz \ldots ay \ldots ax \ldots a$, predicting the token $z$ would constitute a conditional retrieval problem. The retrieval steps in the retrieval problem are perfectly linear, while in the conditional retrieval, they form a directed acyclic graph.

### 3.2 MOTIVATION

The retrieval problem is implicitly present as a subproblem in many common language tasks such as working with relations between persons, tracking the evolution of a concept, solving mathematical and reasoning problems, programming, and many more. Consider the following real-world example from the Wikipedia article on llamas:

*"**Llamas** are social animals and live with others as a **herd**. [ ... ] A **cria** (from Spanish for 'baby') is the name for a baby **llama**, alpaca, vicuña, or guanaco. **Crias** are typically born with all the females of the **herd** gathering around."*

An autoregressive language model trying to predict the second occurrence of the word *herd* (rather than *flock*, *group*, or *pack*) would need to first retrieve the fact that crias are llamas, and then use it to retrieve the fact that llamas live in herds. This process is essentially a retrieval problem with $D = 2$.

From one point of view, the retrieval problem is essentially about working with relations between entities, which is fundamental for language and reasoning. This makes it an ideal testbed for studying the inner workings of large language models.

## 4 LARGE LANGUAGE MODELS

To better illustrate the task and to enable benchmarking of large language models, I propose 5 specific formulations of the retrieval problems: 3 *retrieval* formulations and 2 *conditional retrieval* formulations.

**F1.** *Equations* formulation: "a = 3. b = a. c = b. c = ?"

**F2.** *Lives-with* formulation: "Alice lives in Switzerland. Bob lives with Alice. Charlie lives with Bob. David lives with Charlie. Where does David live?"

**F3.** *Kingdoms* formulation: "Alice lives in Novaria. Novarians believe in harmonianism. Harmonianists eat lamb. Lamb contains Zephyrium. Zephyrium causes Chronogy. Who has Chronogy?"

**F4.** *Functions* formulation (conditional retrieval): "f(2) = 3. g = f. a = 2. g(a) = ?"

**F5.** *Relatives* formulation (conditional retrieval): "Jane lives in Switzerland. Alex's mother is Jane. Engineers live with their mothers. Alex is an engineer. Where does Alex live?"

To ensure that the retrieval problem is not trivially solvable by just finding the noun that fits the question, I interleave multiple retrieval chains in the same question. This ensures that the model performs the entire reasoning chain. To facilitate benchmarking, I also ask the model to output the answer directly without any additional words and I repeat sampling until an acceptable answer is generated. In Appendix A, I provide examples of the full prompts, correct answers, and acceptable answers for each formulation.

I test the large language models on 500 randomly generated questions for each formulation. The results are presented in Figure 2. For the *equations* formulation, I also measure the accuracy for different difficulty levels $D$ (number of equations) and I find that large language models can solve it almost perfectly for $D \leq 5$. Great performance is also achieved on the *lives-with* and *kingdoms* formulations with $D = 5$, as well as on the conditional retrieval formulations *functions* and *relatives*.

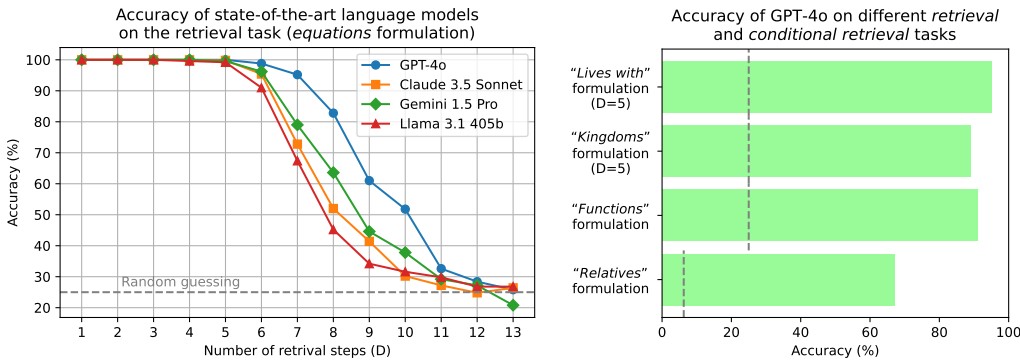

Figure 2: Accuracy of large language models on the retrieval and conditional retrieval problems. Dashed lines indicate the accuracy of random guessing. Full prompts and benchmarking details are provided in Appendix A.

## 5 THEORETICAL ANALYSIS OF INFORMATION FLOW

In this section, I theoretically establish that solving the retrieval problem requires a minimum number of transformer layers that grows logarithmically with the number of retrieval steps $D$. I model the information flow between different positions during self-attention under the following simplifying assumptions:

**Assumption 1.** During self-attention, a position can only attend to another position if they already share a piece of information.

This assumption is motivated by the fact that a position can only attend to another position if their key and query vectors align. Constructing aligned key-query pairs is only possible if the two positions already share some information. More precisely, the shared information must be located in the row spaces of the query and key matrices of the attending and attended positions, respectively.

**Assumption 2.** When a position attends to another position, it retrieves all the information contained in the attended position.

This assumption simplifies the analysis by ignoring network capacity limitations. Solving this setting will give us a lower bound on the number of layers required to solve the retrieval problem in the case of a limited network capacity.

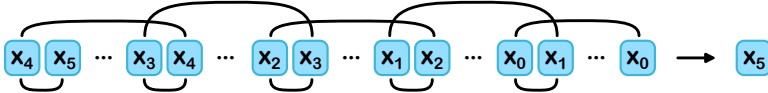

Figure 3: Positions that contain shared information before any transformer layers in the case of $D = 5$. Top edges denote shared token embeddings. Bottom edges denote shared positional encodings.

Based on this simplified model of the attention mechanism, we can prove the following result:

**Theorem 1.** The last position in the sequence cannot retrieve the embedding vector $x_D$ of the target token with $t$ transformer layers if $t < \log_3(2D)$.

This implies that at least $\log_3(2D)$ transformer layers are required to solve a retrieval problem with $D$ retrieval steps. Note that this result is a lower bound that does not take into account the limitations of network capacity, causal masking, or training dynamics. In practice, we can expect the number of required layers to be even higher.

I defer the complete proof to Appendix B. The intuition behind this result comes from the fact that the last position in the sequence cannot retrieve the target token $x_D$ without retrieving everything in-between (i.e., $x_1$, $x_2$, ..., $x_{D-1}$). However, it is possible to show that the number of retrieved tokens can grow at most by a factor of 3 after each attention layer, hence the logarithmic growth of the number of required layers.

## 6 MINIMAL PROBLEM FORMULATION

In order to better study the mechanism by which transformers solve the retrieval problem, I introduce a minimal formulation of the retrieval problem with $N$ retrieval chains, $D$ retrieval steps per chain, and $K$ embedding dimensions. I use $N = 4$ and $K = 4$ throughout. Each chain contains $D + 1$ unique symbols forming $D$ pairs and one query. For every input sequence, each of the $N(D + 1)$ unique symbols is assigned a $K$-dimensional vector whose components are sampled i.i.d from a standard normal distribution.

I create the input sequences by perfectly interleaving the pairs of symbols forming each retrieval chain, followed by the $N$ query symbols. I randomly shuffle the query vectors. I also shuffle the input pairs from different chains within the same retrieval step. Finally, I concatenate each token embedding with a $K$-dimensional rotary positional encoding (Su et al., 2023). Each input sequence will contain $N(2D + 1)$ vectors of dimension $2K$. The output sequences consist of $N$ vectors, one for each query token.

## 7 IMPLICIT CURRICULUM & NUMBER OF LAYERS

I consider two possible formulations: an implicit curriculum (IC) formulation and a non-IC formulation. In the IC formulation, the target vectors have $DN$ dimensions and contain all the tokens forming each retrieval chain concatenated (except the query token $x_0$). In the non-IC formulation, the target vectors are $K$-dimensional and contain only the last token of each retrieval chain, namely $x_D$.

My initial experiments suggest that the implicit curriculum (IC) plays a crucial role in the successful learning of the retrieval problem. To better quantify this effect, I conduct two comprehensive sets of experiments, one for each formulation (IC and non-IC). For each formulation, I train 64 transformers with 1 to 8 layers (8 transformers for each number of layers). I plot the final validation loss averaged across all runs with the same number of layers in Figure 4 (left).

To better understand the connection between the number of layers and the difficulty of the retrieval problem, I also plot the partial validation loss for each position in the retrieval chains in the IC formulation (Figure 4, right). I use $D = 5$ for the IC formulation to better illustrate this connection, but only $D = 3$ for the non-IC formulation to illustrate the importance of the implicit curriculum.

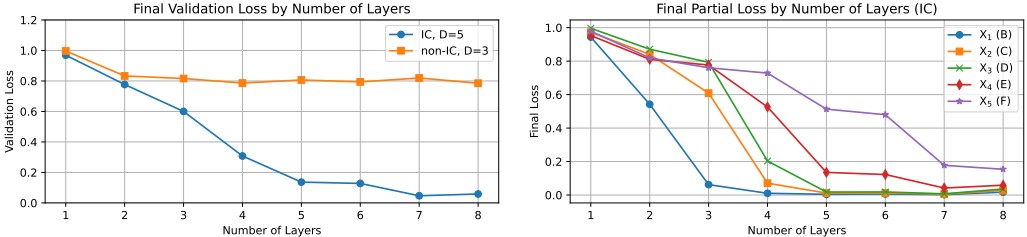

Figure 4: Final validation loss by number of layers, averaged across multiple runs. Left: IC *vs.* non-IC formulations. Right: Partial validation loss for each position in the retrieval chains (IC only).

## 7.1 TRAINING DETAILS

For each formulation and number of layers, I train 8 transformers following the recipe of Radford et al. (2019). Each transformer has 8 attention heads per layer and residual streams of size 128. I train for 10k steps using the Adam optimizer (Kingma, 2014) with a learning rate of $10^{-3}$, decoupled weight decay of $0.1$ (Loshchilov, 2017), a batch size of 512, $2^{20}$ randomly generated training examples, layer normalization (Ba et al., 2016), no dropout, and mean squared error loss. We measure the final validation loss by averaging the validation loss over the last 100 training steps.

## 7.2 RESULTS

First, I observe that the IC formulation is essential for successful learning. In the non-IC formulation, the transformers fail to learn the retrieval problem even for $D = 3$, regardless of the number of layers. I confirm that for 100% of the non-IC runs, the final validation loss is above $0.7$.

Second, I empirically confirm the connection between the number of layers and the difficulty of the retrieval problem. For the IC formulation, the later positions in the retrieval chains (corresponding to greater $D$) are more difficult to learn and require more layers.

Third, I find our first hint regarding the emergence of retrieval heads. During training with IC, the partial losses for earlier positions in the retrieval chains always decrease faster than the partial losses for later positions. I confirm that in 100% of the IC runs, the partial loss goes below $0.5$ for $x_1$ first, then for $x_2$, and so on. I will further investigate this phenomenon in section 9.

## 8 REVERSE-ENGINEERING THE CIRCUITS LEARNED

To understand the mechanism learned by transformers to solve the retrieval problem, I train three transformers (denoted as A, B, and C) with 12 layers and only one attention head per layer on the retrieval problem with $D = 3$. I then manually reverse-engineer the circuits learned by the transformers by studying their attention maps. The uncovered circuits are depicted in Figure 5. I describe my reverse-engineering process in detail in Appendix E.

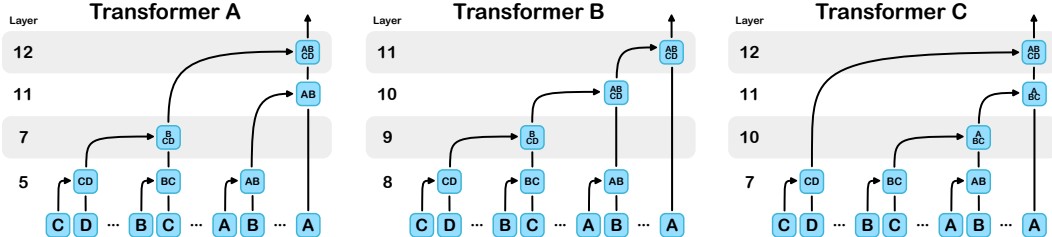

Figure 5: Reverse-engineered circuits from three 12-layer transformers trained on the retrieval problem with $D = 3$ and IC.

## 8.1 TRAINING DETAILS

I follow a similar training recipe as in the previous section. Each transformer has 12 layers, one attention head per layer, and residual streams of size 128. I train each transformer for 24k steps, a batch size of 128, and 262k randomly generated training examples (IC).

## 8.2 RESULTS

I find that transformers A, B, and C achieve a validation mean squared error of less than $0.01$. By studying the attention maps in the trained transformers, I observe that most attention heads do not perform any useful computation. Only a few attention heads are responsible for the information flow. Their behavior is easily interpretable (see Appendix C).

I manually reverse-engineer the entire circuits learned by the three transformers, which are depicted in Figure 5. I perform extensive validations of the circuits using ablations. My reverse-engineering process is described in detail in Appendix E.

I observe two interesting facts about the reverse-engineered circuits:

**i.** First, in all three transformers, the first relevant attention head is connecting the first and second tokens in each input pair, enabling the subsequent attention heads to attend to the second token in the pair using the value of the first token. This mechanism is highly reminiscent of the induction head mechanism (Olsson et al., 2022; Reddy, 2023).

**ii.** Second, except for the first attention head, the circuits learned by the transformers are very different. Interestingly, none of the transformers use the minimum number of attention heads required to solve the retrieval problem for $D = 3$. All transformers use 4 attention heads, but it is possible to use only 3 (for example, by combining layers 7 and 11 in transformer A).

# 9 EMERGENCE OF ATTENTION HEADS DURING TRAINING

To better understand how the retrieval heads emerge during training, I train a 24-layer transformer (denoted as Transformer D) on the retrieval problem with $D = 4$ and IC. I manually reverse-engineer the circuits learned by Transformer D following the same procedure described in Appendix E. Afterward, I measure the attention during training for each attention path in the reverse-engineered circuit.

## 9.1 TRAINING DETAILS

I follow a similar training recipe as in the previous sections. Transformer D has 24 layers, one attention head per layer, and residual streams of size 512. I train for 6400 steps (800 epochs), a batch size of 256, and 262k randomly generated training examples (IC, $D = 4$). To speed up the training and reduce the checkpoint size, I remove the MLPs and reduce the head size to 16.

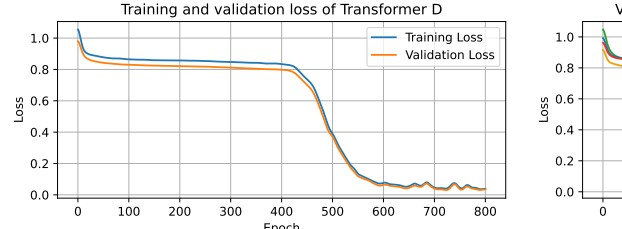 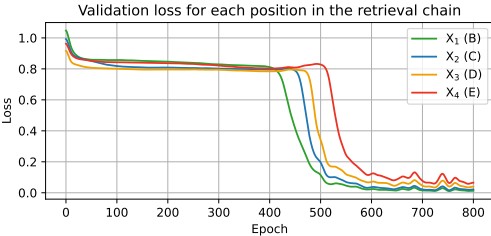

Figure 6: Loss during training of Transformer D (24 layers) with IC and $D = 4$. Left: training and validation loss. Right: partial validation loss for each position in the retrieval chain.

I save a checkpoint every 10 epochs (80 steps) that I later use to measure the average attention for each attention path in the reverse-engineered circuit, at each epoch during training, using 32 input sequences.

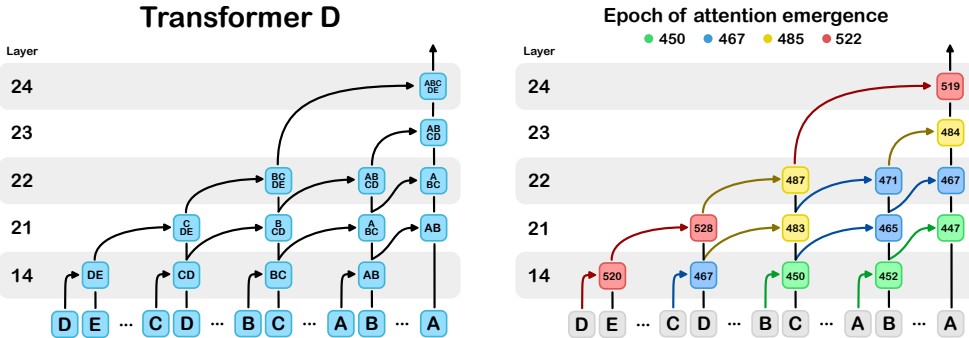

Figure 7: Left: Reverse-engineered circuits of Transformer D. Right: the epoch when the average attention goes above $0.5$ for each attention path.

## 9.2 RESULTS

Transformer D achieves a validation mean squared error of $0.031$. I plot the training, validation, and partial validation loss in Figure 6. Using the same reverse-engineering procedure, I uncover a more complex circuit than before, with multiple paths connecting the same positions (Figure 7, left). After ablation, the mean squared error increases to $0.045$.

For every checkpoint, I measure the average attention for each attention path in the reverse-engineered circuit. I approximate the attention between checkpoints using linear interpolation. I display the plots for each attention path in Appendix D. Finally, I show the first epoch when the average attention goes above $0.5$ for each attention path (Figure 7, right).

By analyzing the partial loss curves and the emergence of attention paths, we can make the following observations:

    **i.** After 450 epochs of slow learning, an induction head that can retrieve token **B** emerges abruptly on layers 14 and 21. This drives down the first partial loss.

   **ii.** Quickly after, another attention head emerges on layer 22. This head reuses the induction head (with slight adjustments) to retrieve token **C** and drive down the second partial loss.

  **iii.** Finally, two more heads emerge on layers 23 and 24 that reuse heads 14, 21, and 22 to retrieve tokens **D** and **E**, respectively. This drives down the last two partial losses.

  **iv.** Head 24 emerges much later than head 23, possibly due to the greater modifications required to reuse the existing circuit.

Together, these observations strongly suggest the following possible explanation for the importance of the implicit curriculum: *The implicit curriculum provides a sequence of increasingly complex tasks that enables learning the entire retrieval mechanism one head at a time, starting with an induction head.*

## 10    DISCUSSION ON EMERGENT ABILITIES

The retrieval problem has interesting connections to the emergent abilities of large language models (Wei et al., 2022). An ability is *emergent* if it is not present in smaller models but is present in larger models. Understanding emergence is an important direction because it could potentially allow us to predict what abilities future models may have, as well as provide new insights into how to train more capable language models.

The existence of tasks that require a minimum number of layers, such as the retrieval problem, provides a possible explanation for the emergence of new abilities in large language models. As the model grows in size, it becomes possible to learn more complex circuits that would be impossible to learn in smaller models. This unlocks new abilities that were previously unattainable.

### 10.1    EMERGENT ABILITIES ARE NOT A MIRAGE

Schaeffer et al. (2023) have previously suggested that the emergence of new abilities in large language models is just a "mirage" that appears only under nonlinear or discontinuous metrics. My work provides a very strong counterargument to this claim if we consider the ability of a model to solve the retrieval problem. A transformer cannot solve the retrieval problem with a specific difficulty unless it has the minimum number of necessary layers.

### 10.2    THE IMPLICIT CURRICULUM OF NATURAL LANGUAGE

Is it possible to understand how training on natural language data leads to reasoning abilities? Chan et al. (2022) previously found that natural language has specific data distributional properties that enable emergent in-context learning in transformers. Is it possible to achieve a similarly insightful understanding of the emergence of reasoning abilities in general, not just in-context learning?

The retrieval problem provides a promising avenue for answering this question. In Sections 7 and 9, we saw that retrieval heads can only emerge gradually, one by one, under the presence of an implicit curriculum that provides a sequence of increasingly complex tasks. This suggests that one property of natural language data is incredibly important for the emergence of reasoning abilities: the presence of a very diverse set of tasks with varying levels of difficulty.

## 11    RELATED WORK

**Single-layer transformers.**    Perhaps the most well-studied setting for single-layer transformers is the problem of modular addition. Nanda et al. (2023) show that transformers solve modular addition by arranging the embedding vectors in a circular structure and leveraging the attention mechanism to perform trigonometric operations. Zhong et al. (2024) extend this work by uncovering other algorithms and embedding structures. Even the training dynamics are beginning to be understood, with Ding et al. (2024) studying the survival of initial circular representations and my previous work (Musat, 2024) proposing an effective theory of the training dynamics by modeling the embeddings as a particle system. Quirke et al. (2023) train a single-layer transformer with three attention heads on the problem of n-digit integer addition. They find that transformers break down the multi-digit addition task into parallel, digit-specific streams, using different algorithms for various digit positions.

**Two-layer transformers.**    By studying two-layer transformers, Olsson et al. (2022) uncover a mechanism termed *induction head* that, given an input sequence $ab \ldots a$, can predict $b$. One possible use of an induction head is *sequence copying*, but the authors argue that it can also perform more high-level functions such as *translation*. The induction head is formed by two stacked attention heads. The first head copies into the residual stream of $b$ the value of the previous token $a$. The

second head is then able to attend to the token $b$ and copy it into the residual stream of the final token. Reddy (2023) explains the emergence of the induction head during training by the sequential learning of three nested logits enabled by an implicit curriculum.

**Large language models.** Several studies on large language models use automated or semi-automated methods to isolate circuits that solve a specific task (Conmy et al., 2023; Goldowsky-Dill et al., 2023). Such circuits often span many layers, but their mechanisms remain not fully understood (Wang et al., 2022). Attention heads in large language models are often strongly inter-dependent, which makes it difficult to isolate and understand individual heads (Bricken et al., 2023). In large language models, even the simple task of greater-than comparison, which could in principle be solved by a single-layer transformer, is solved by a complex mechanism formed by multiple attention heads and MLPs (Hanna et al., 2024).

## 12    CONCLUSION

In this work, I introduced and studied the retrieval problem, a simple task that requires transformers to retrieve information from multiple positions in the input sequence. I showed that the retrieval problem requires a certain number of layers to be solved. By training transformers on a minimal formulation of the task, I found that transformers solve the task using a mechanism that resembles an induction head. I found that this mechanism emerges gradually with the help of an implicit curriculum, starting with an induction head and then adding more heads one by one.

**Limitations.** My analysis of transformers trained on a minimal formulation of the retrieval problem might not generalize perfectly to large language models. I also do not provide a full explanation of the training dynamics of transformers on the retrieval problem. Further research is needed to fully understand the multi-layered circuits learned by large language models and the training dynamics that enable their learning.

**Acknowledgments.** I would like to thank the anonymous reviewers for their valuable feedback.

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

## A  RETRIEVAL AND CONDITIONAL RETRIEVAL PROMPTS

Below I provide one complete example for each of the retrieval and conditional retrieval formulations used in the paper. The examples are generated using the same programs used for benchmarking the large language models. Each example consists of a prompt, a correct answer, and acceptable answers. Acceptable answers are used to filter out incoherent answers by repeatedly sampling from the model until an acceptable answer is found.

### A.1  EQUATIONS FORMULATION ($D = 5$)

```
b  =  2
c  =  3
d  =  0
a  =  1
e  =  b
g  =  a
h  =  d
f  =  c
k  =  e
i  =  f
l  =  g
j  =  h
n  =  k
p  =  l
o  =  i
m  =  j
q  =  n
s  =  p
r  =  o
t  =  m
What  is  the  value  of  s?  Say  directly  only  the  numeric  value,
     without  any  other  words.

Correct:  1
Acceptable:  0,  1,  2,  3
```

### A.2  LIVES-WITH FORMULATION ($D = 5$)

```
Charlie  lives  in  Cairo
David  lives  in  Delhi
Alice  lives  in  Berlin
Bob  lives  in  Amsterdam
Henry  lives  with  David
Eve  lives  with  Charlie
Frank  lives  with  Alice
Grace  lives  with  Bob
Kate  lives  with  Grace
Larry  lives  with  Frank
Jack  lives  with  Eve
Isabelle  lives  with  Henry
Mary  lives  with  Jack
Olivia  lives  with  Isabelle
Nick  lives  with  Kate
Peter  lives  with  Larry
Rose  lives  with  Peter
Queen  lives  with  Nick
Tom  lives  with  Olivia
Sam  lives  with  Mary
```

Where does Rose live? Say directly only the name of the city, without any other words.

Correct: Berlin
Acceptable: Amsterdam, Berlin, Cairo, Delhi

### A.3 KINGDOMS FORMULATION

Bob lives in Silvania.
Alice lives in Novaria.
Charlie lives in Aurora.
David lives in Florinia.
Silvanians believe in celestianism.
Novarians believe in harmonianism.
Aurorans believe in elysianism.
Florinians believe in luminism.
Luminists eat beef.
Elysianists eat pork.
Harmonianists eat lamb.
Celestianists eat chicken.
Beef contains Astralyte.
Chicken contains Nephryon.
Lamb contains Zephyrium.
Pork contains Virellium.
Zephyrium causes Chronogy.
Astralyte causes Aetherflux.
Virellium causes Somnosis.
Nephryon causes Synthemia.
Who has Chronogy? Say directly the name without other words.

Correct: Alice
Acceptable: Alice, Bob, Charlie, David

### A.4 FUNCTIONS FORMULATION (CONDITIONAL RETRIEVAL)

a(0) = 3
a(1) = 2
a(2) = 0
a(3) = 1
b(0) = 1
b(1) = 3
b(2) = 2
b(3) = 0
c(0) = 1
c(1) = 0
c(2) = 3
c(3) = 2
d(0) = 1
d(1) = 0
d(2) = 2
d(3) = 3
e = b
f = a
g = c
h = d
i = 0
j = 2
k = 3

```
l = 1
```
What is the value of f(i)? Say directly only the numeric value,
    without any other words.

Correct: 3
Acceptable: 0, 1, 2, 3

### A.5 RELATIVES FORMULATION (CONDITIONAL RETRIEVAL)

Penny lives in Canada.
Lily lives in Brazil.
Isabelle lives in France.
Cathy lives in Kenya.
George lives in Italy.
Adam lives in Mexico.
Kevin lives in Peru.
Ed lives in Laos.
Hank lives in Germany.
Mike lives in Japan.
Jane lives in England.
Fred lives in Hungary.
Dana lives in Norway.
Olivia lives in Qatar.
Bob lives in Denmark.
Nancy lives in Argentina.
John's mother is Jane.
John's sister is Olivia.
John's father is Ed.
John's brother is Mike.
Chris's mother is Penny.
Chris's sister is Dana.
Chris's father is Adam.
Chris's brother is George.
Diana's mother is Nancy.
Diana's sister is Isabelle.
Diana's father is Hank.
Diana's brother is Bob.
Eve's mother is Lily.
Eve's sister is Cathy.
Eve's father is Fred.
Eve's brother is Kevin.
Doctors live with their brothers.
Lawyers live with their mothers.
Teachers live with their sisters.
Engineers live with their fathers.
John works as a doctor.
Chris works as an engineer.
Diana works as a teacher.
Eve works as a lawyer.
Where does Eve live? Say directly only the name, without any other
    words.

Correct: Brazil
Acceptable: Argentina, Brazil, Canada, Denmark, England, France,
    Germany, Hungary, Italy, Japan, Kenya, Laos, Mexico, Norway,
    Peru, Qatar

## B    PROOF OF THEOREM 1 (MINIMUM NUMBER OF LAYERS)

In this section, I provide the complete formal proof for **Theorem 1** stated in Section 5. Recall the two assumptions underlying our simplified model of the attention mechanism:

**Assumption 1.** During self-attention, a position can only attend to another position if they already share a piece of information.

**Assumption 2.** When a position attends to another position, it retrieves all the information contained in the attended position.

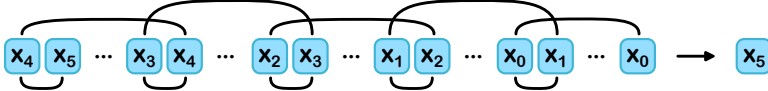

Figure 8: Positions that contain shared information before any transformer layers in the case of $D = 5$. Top edges denote shared token embeddings. Bottom edges denote shared positional encodings.

**Definition 1.** Let's consider all relevant input positions ordered by their reachability from the last token following the paths of shared information before any attention layers, exactly as depicted in Figure 8 (e.g., $\boldsymbol{x}_0\boldsymbol{x}_0\boldsymbol{x}_1\boldsymbol{x}_1\boldsymbol{x}_2\boldsymbol{x}_2\ldots$ ). Let's denote this sequence of positions as $\boldsymbol{s}_i$ for $i \in \{0, 1, \ldots, 2D\}$. In other words, $\boldsymbol{s}_{2k}$ and $\boldsymbol{s}_{2k+1}$ are the positions of the second and first occurrences of $\boldsymbol{x}_k$, respectively. For example, $\boldsymbol{s}_0$ is the last position in the input sequence, $\boldsymbol{s}_1$ is the first occurrence of $\boldsymbol{x}_0$, $\boldsymbol{s}_2$ is the second occurrence of $\boldsymbol{x}_1$, $\boldsymbol{s}_{2D}$ is the position of the target token $\boldsymbol{x}_D$, and so on.

**Definition 2.** Let's denote as $\boldsymbol{r}_{t,i}$ the residual stream at postition $\boldsymbol{s}_i$ after layer $t$. Before any transformer layers, the residual stream contains only the token embedding and positional enconding:

$$\boldsymbol{r}_{0,i} = \boldsymbol{x}_{\lfloor i/2 \rfloor} + \boldsymbol{p}_{\lfloor (i+1)/2 \rfloor} \quad \text{for all } i \in \{0, 1, \ldots, 2D\}, \tag{1}$$

where $\lfloor \cdot \rfloor$ denotes the floor function, $\boldsymbol{x}_k$ is the $k$-th token embedding, and $\boldsymbol{p}_k$ is the positional encoding of the $k$-th input pair.

**Lemma 1.** During self-attention, a position can only attend to another position if they already contain a shared token embedding or positional encoding.

*Proof.* In the retrieval problem, the token values and pair positions are assigned randomly and independently. Knowledge of a token or position does not provide any information about any other token or position. Therefore, the only way for two positions to share information is if they already contain a shared token embedding or positional encoding.

Note that before any attention layers, the position $\boldsymbol{s}_i$ will contain information shared only with positions $\boldsymbol{s}_{i-1}$ and $\boldsymbol{s}_{i+1}$. This is because the token embedding $\boldsymbol{x}_i$ is shared by $\boldsymbol{s}_{2i}$ and $\boldsymbol{s}_{2i+1}$, while the positional encoding $\boldsymbol{p}_i$ (for $i$-th input pair) is shared by $\boldsymbol{s}_{2i-1}$ and $\boldsymbol{s}_{2i}$.

**Definition 3.** Let's denote as $\boldsymbol{e}_i$ the piece of information shared by the positions $\boldsymbol{s}_i$ and $\boldsymbol{s}_{i+1}$ before any transformer layers. Specifically, we define $\boldsymbol{e}_i$ for all $i \in \{0, \ldots, 2D\}$ such that $\boldsymbol{e}_{2k} = \boldsymbol{x}_k$ for $k \in \{0, 1, \ldots, D\}$ and $\boldsymbol{e}_{2k-1} = \boldsymbol{p}_k$ for $k \in \{1, \ldots, D\}$.

We are interested in the minimum number of layers $t$ such that $\boldsymbol{r}_{t,0}$ might contain the target token $\boldsymbol{x}_D$, also denoted as $\boldsymbol{e}_{2D}$.

**Lemma 2.** After every layer, every residual stream $\boldsymbol{r}_{t,i}$ will contain a contiguous sequence of pieces of information (e.g., $\{\boldsymbol{e}_a, \boldsymbol{e}_{a+1}, \ldots, \boldsymbol{e}_b\}$).

*Proof.* We can show this using mathematical induction. The initial residual stream $\boldsymbol{r}_{0,i}$ contains only the token embedding and the positional encoding, which represent the consecutive pieces of information $\{\boldsymbol{e}_i, \boldsymbol{e}_{i+1}\}$. During self-attention, the existing contiguous sequence of pieces of information in $\boldsymbol{r}_{t,i}$ will be merged with other contiguous sequences (assumption 2) that share at least one piece of information with $\boldsymbol{r}_{t,i}$ (lemma 1). Their union in $\boldsymbol{r}_{t+1,i}$ remains a contiguous sequence.

**Lemma 3.** After every layer $t$, the contiguous sequence of pieces of information in $\boldsymbol{r}_{t,i}$ will have a length of at most $3^t + 1$ for all $i$.

*Proof.* We can show this using mathematical induction once again. The initial residual streams $r_{0,i}$ contain exactly two pieces of information: the token embedding and the positional encoding. During the $t$-th layer of self-attention, the contiguous sequence of pieces of information in $r_{t-1,i}$ will grow by at most $3^{t-1}$ pieces of information to the left and the right, resulting in a new total length of at most $3^t + 1$.

**Theorem 1.** The embedding vector $x_D$ of the target token cannot be present in the residual stream $r_{t,0}$ after $t$ layers if $t < \log_3(2D)$.

*Proof.* The embedding of the target token $x_D$ corresponds to the piece of information $e_{2D}$. For $r_{t,0}$ to contain $e_{2D}$, the length of its contiguous sequence must be at least $2D + 1$. By Lemma 3, this length will not be reached if $3^t + 1 < 2D + 1$, which is equivalent to $t < \log_3(2D)$.

# C    ATTENTION MAPS

## C.1    TRANSFORMER A

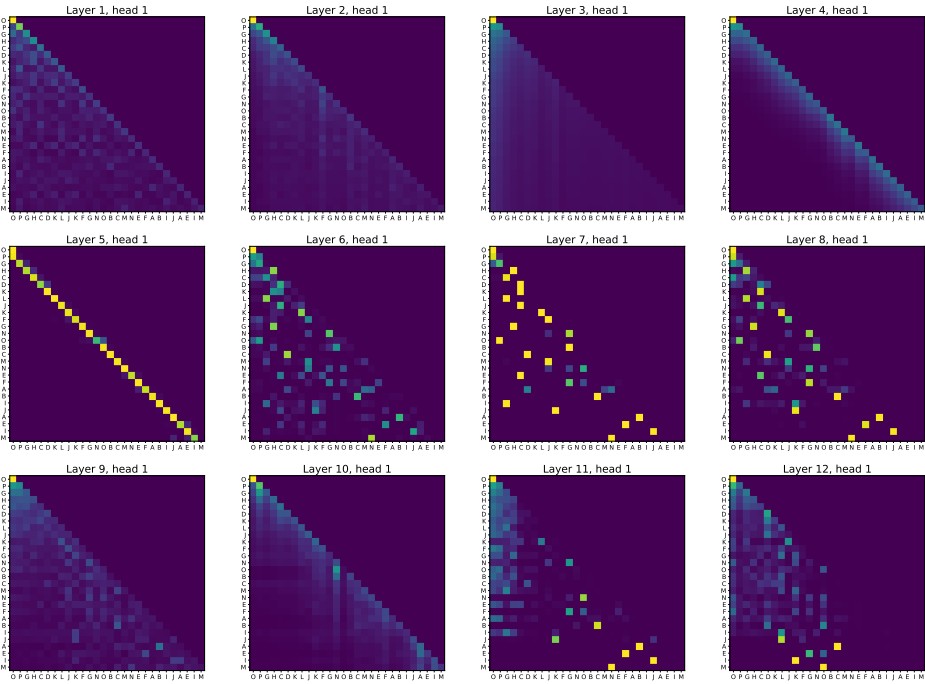

## C.2    TRANSFORMER B

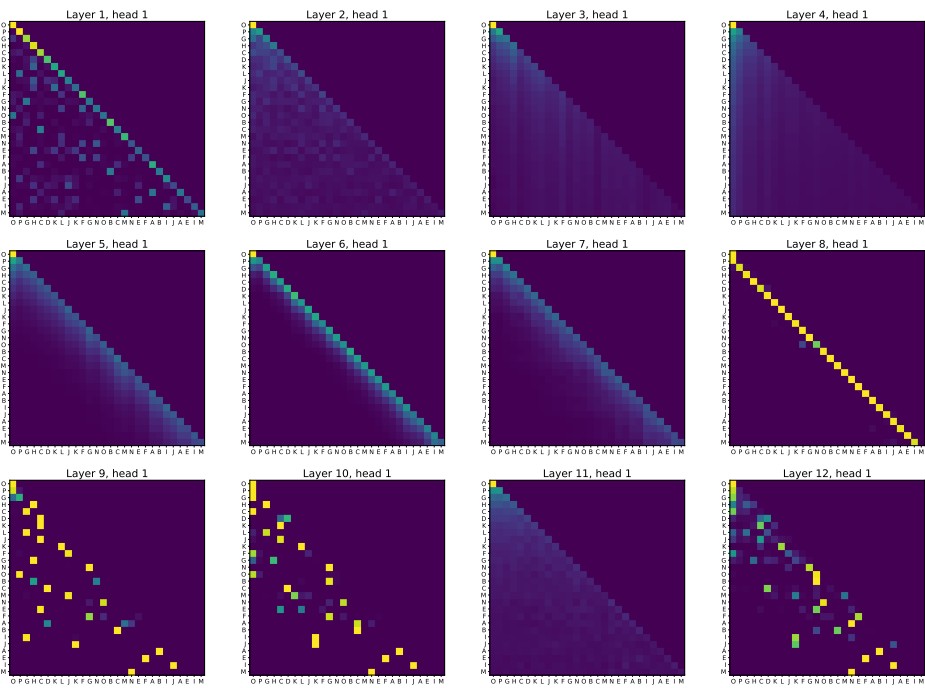

## C.3 TRANSFORMER C

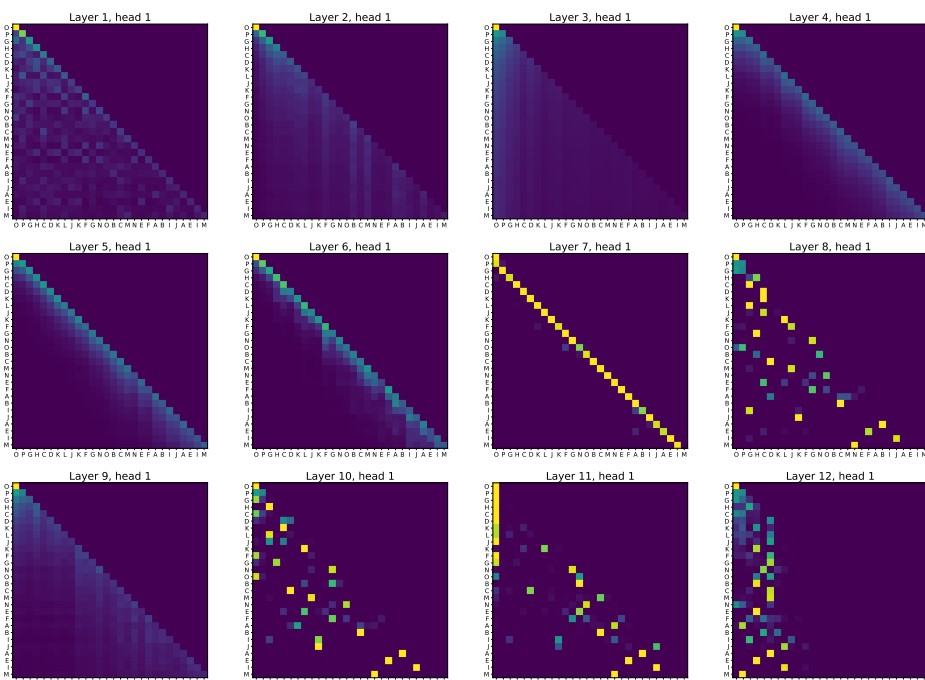

## C.4 TRANSFORMER D

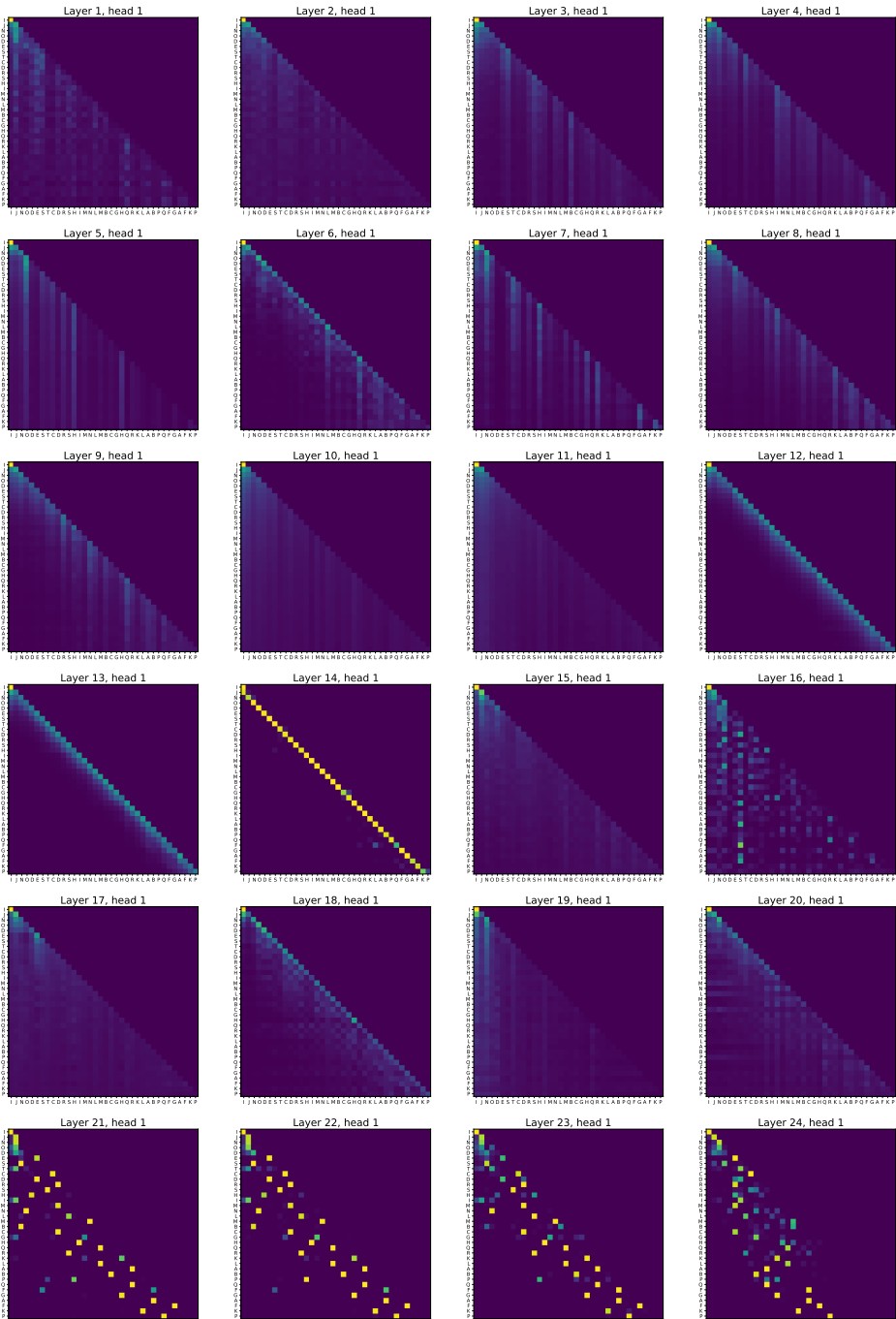

# D ATTENTION EMERGENCE IN TRANSFORMER D

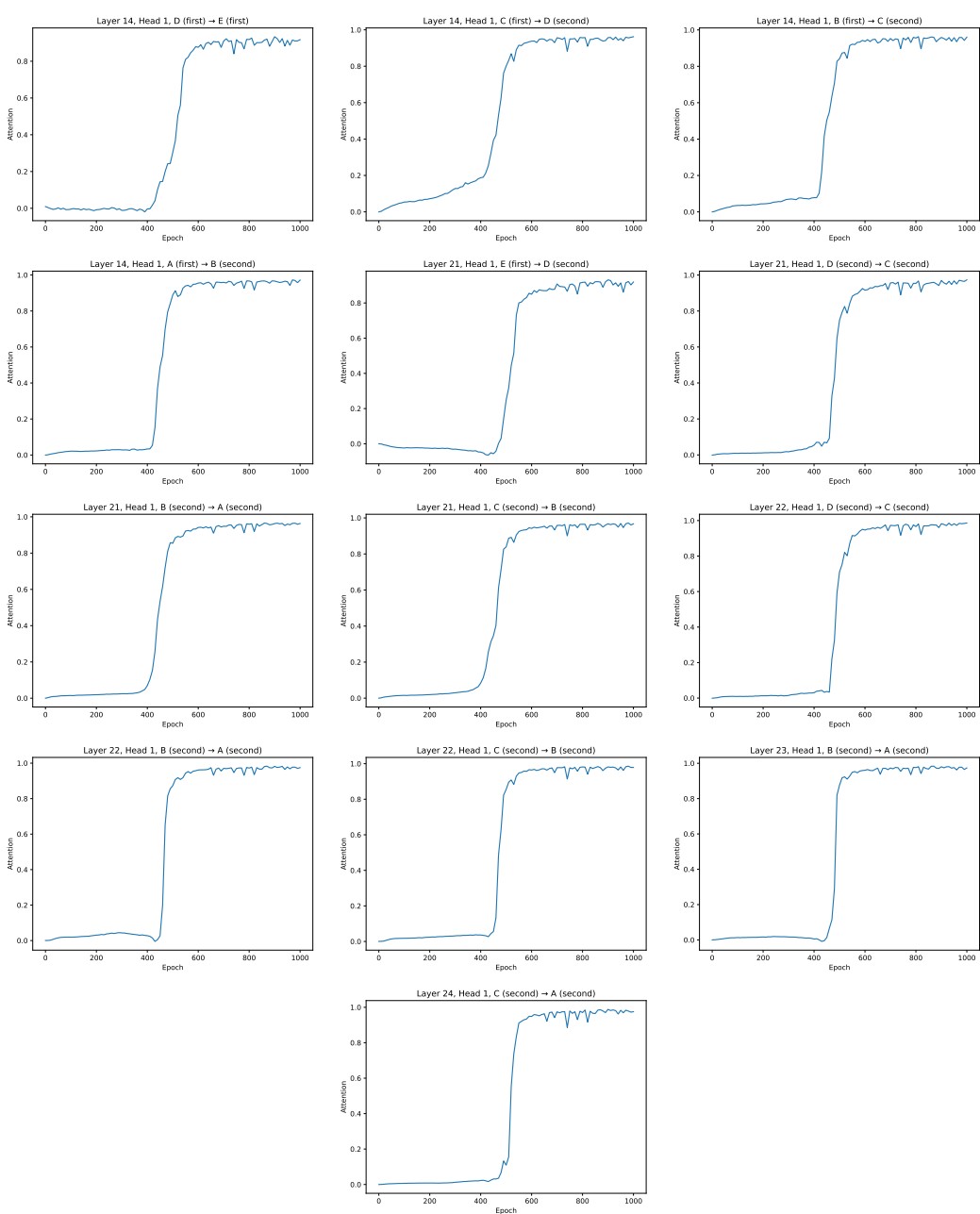

# E  PROCEDURE FOR REVERSE-ENGINEERING CIRCUITS

As can be seen in Appendix C, the attention maps for each attention head in the transformers contain clear patterns that can be be manually identified without the need for any additional tools. For this reason, I decided to manually reverse-engineer the circuits, while validating them thoroughly using ablations to ensure their correctness.

The exact procedure I follow to reverse-engineer the circuits:

1. I plot the attention maps for each transformer and each layer for different prompts.
2. By observing the attention maps, I identify several possible mechanisms that could explain the attention patterns of each head.
3. For each head, I determine which of the hypothesized mechanisms is correct using ablations (described below) and measuring the validation loss. I choose the simplest mechanism that maintains a low validation loss (below $0.05$) after ablation.
4. I repeat steps 1-3 until the mechanism of each head has been identified.
5. I validate the complete mechanism by performing combined ablations on all heads and measuring the validation loss.
6. I validate that the uncovered mechanism is not excessive by attempting to further ablate all attention paths individually and measuring the validation loss.

To validate the circuits, I measure the validation error after ablating the attention maps in the following manner. For the attention heads that do not perform any useful computation, I replace the attention weights with either uniform attention or an identity matrix. For the attention heads that are responsible for the information flow, I construct an attention map that is zero everywhere except for the position that I expect the head to attend to, where it is set to one.

After performing the combined ablations (step 5), I find that the mean squared error increases slightly, but remains below $0.05$ for all transformers. By further ablating any apparently useful attention path (step 6), the mean squared error increases to $0.17 - 0.89$. The only exceptions are the first useful layers of each transformer, which always attend to the previous position. After ablating their attention as uniform, the error stays in the range $0.05 - 0.1$, suggesting that they do not contribute directly to the final output, but rather enable the information flow in the subsequent layers.

