# OpenReview forum: "Mechanism and Emergence of Stacked Attention Heads in Multi-Layer Transformers"
_ICLR.cc/2025/Conference — ICLR 2025 Poster_

### Official Review · Reviewer_yct3 · 2024-11-03

**Soundness:** 2
**Presentation:** 2
**Contribution:** 2
**Rating:** 5
**Confidence:** 3

**Summary:**

This paper introduces the retrieval problem, a fundamental task that challenges transformers to retrieve information from multiple positions within an input sequence. The authors demonstrate that solving the retrieval problem requires a certain depth of transformer layers.
 This paper explores how large language models and transformers handle retrieval tasks. They introduce two tasks: the retrieval problem and a variant called the conditional retrieval problem. It shows that large language models can solve retrieval tasks without fine-tuning by leveraging complex mechanisms involving multiple attention heads, which emerge in a specific sequence when trained with an implicit curriculum.

**Strengths:**

● The paper provides insights into how LLMs perform retrieval using attention heads.
● Introducing these tasks gives a clear way to study transformers' retrieval abilities.
● The study highlights the role of learning curriculum in the development of retrieval mechanisms.

**Weaknesses:**

1. **Lack of Experimental Validation for Theoretical Claims**: Theoretical claims like Theorem 1 lack empirical support, making it hard to verify their practical impact.
2. **Insufficient Details on Experimental Setup**:: The paper lacks detailed explanations for key experimental settings. The implicit curriculum (IC) formulation, which performs better than non-IC, is not clearly defined, and Section 8’s description of manually reverse-engineering circuits lacks detail, making the experiments difficult to reproduce.

**Questions:**

1. Assumption 1, which requires shared positional or token information for attention between positions, is overly restrictive and conflicts with the flexibility of Transformer models. Transformers are designed to allow any position to attend to any other, enabling them to learn complex relationships without pre-shared information. This assumption limits the model's ability to capture long-range dependencies and generalize to tasks where connections are context-driven rather than based on shared information. I recommend reconsidering this assumption to better align with the strengths of the Transformer architecture.
2. Theorem 1  would benefit from experimental validation. Testing whether the target embedding truly appears in the residual stream only when \( t \geq \log_3(2D) \) would strengthen the claim and confirm its practical relevance. I recommend adding experiments to verify this behavior across different settings.
3. The specific details of the implicit curriculum formulation remain vague, and the reasons for its superior performance are not clearly explained. Additionally, it would be helpful to clarify why the non-IC formulation, which aligns more closely with traditional problem setups, fails to perform well.
4. In Section 8,  "manually reverse-engineering the circuits learned by the transformers" , the specific details of this process are not fully explained. Providing more information on the methodology used to reverse-engineer these circuits, as well as any criteria or steps involved, would enhance the clarity and reproducibility of this analysis.
5. The experimental results need to be validated under more complex settings, such as with longer retrieval steps.
We will increase the score based on the answer to the question.

---

> ### Author Response · Authors · 2024-11-16
>
> Thank you very much for your review! We provide answers to your questions below.
> >Assumption 1, which requires shared positional or token information for attention between positions, is overly restrictive and conflicts with the flexibility of Transformer models.
>
> To be clear, we do not believe that Assumption 1 would hold for transformers trained on any task. However, Assumption 1 **does hold** for transformers trained on the retrieval problem.
>
> During self-attention, a position A can attend to another position B only if A's query vector aligns with B's key vector. This key-query alignment is only possible if the residual streams of A and B already share some information. More precisely, the shared information is located in the row spaces of the Query and Key matrices of A and B, respectively. Without this, it is impossible for A and B to construct aligned key and query vectors. Note that we are using "shared information" in a very abstract sense. This shared information could be a token embedding, a positional encoding, or any nonlinear combination or transformation thereof. In the context of the retrieval problem, we only consider pure token embeddings and positional econdings since they cannot be meaningfully combined or transformed to obtain "new shared information". This is why Assumption 1 is referring only to token embeddings and positional econdings specifically.
>
> We are confident that it is possible to derive a better proof starting from a more realistic transformer model and weaker assumptions. In fact, we think it might even be possible to have a proof that doesn't use any assumptions or simplifications. We have very promising preliminary results in this direction. However, such a proof would also be much longer and more complicated. For the sake of this paper, we believe that the simplified proof provides sufficient support for the task without taking too much space from the other important sections.
> >Theorem 1 would benefit from experimental validation
>
> We already do this to some extent in Section 7, Figure 4 (right). The section is not focused on explicitly proving Theorem 1, but it can also be interpreted in this way. I will provide a comprehensive analysis to make the argument more convincing.
>
> It is particularly important to know that losses of 0.8 are possible without actually solving the problem. You can notice this from figure 4 (left) and figure 7.
>
> Here is a comprehensive analysis of Figure 4 (right):
> - 1 layer: impossible to solve
> - 2 layers: solved for D = 1, remains unsolved for D > 1
> - 3 layers: solved for D = 2, remains unsolved for D > 2
> - 4 layers: solved for D = 3,4,5
>
> We agree that it would be interesting to perform some experiments to provide more clear empirical evidence for theorem 1, but we would like not to remove any of the existing experiments. What do you think about adding an appendix with empirical evidence for Theorem 1?
>
> >The specific details of the implicit curriculum formulation remain vague...
>
> The implicit curriculum (IC) formulation is described on lines 235-237. In the IC formulation, the target vectors have DK dimensions and contain all the tokens forming the retrieval chain concatenated (except the query token x0). For example, for the input sequence "CD…BC…AB…A", the IC formulation would predict the concatenation of [B, C, D].
>
> Regarding learning, our intuition is that it is impossible for gradient descent to learn mechanisms of more than 3 transformer layers. Even 2 layer mechanisms are very hard to learn, as shown by Reddy (2023). As we say in lines 428-431, the implicit curriculum provides a sequence of increasingly complex tasks that enables learning the entire retrieval mechanism one head at a time, starting with an induction head.
>
> >In Section 8, "manually reverse-engineering the circuits learned by the transformers" , the specific details of this process are not fully explained
>
> Thank you, we will add more information on the methodology. Our process is quite standard:
>
> 1. We plotted attention maps for each head.
> 2. We created a list of hypothesized mechanisms for each head based on the attention maps.
> 3. We found the correct mechanism for each head using ablations.
> 4. We validated the complete mechanism by performing all ablations together.
> 5. We validated that our mechanism is not excessive by attempting to perform further ablations.
>
>
> > The experimental results need to be validated under more complex settings, such as with longer retrieval steps.
>
> Can you be more specific? For the experiments in section 7, we use 5 retrieval steps, which we thought to be a good number since it is the maximum that LLMs can solve perfectly. For sections 8 and 9 we decided to go with just 3 and 4 retrieval steps because our focus is on understanding and illustration. Our goal is to show the community what kind of mechanisms become possible with more than 2 transformer layers. If you have a specific experiment in mind, we would be happy to include it in an Appendix.

---

> ### Author Response · Authors · 2024-11-30
>
> As the discussion period is soon drawing to a close, we would greatly appreciate if you could provide a response to our rebuttal and updated paper. Thank you!

---

### Official Review · Reviewer_cnTJ · 2024-11-03

**Soundness:** 3
**Presentation:** 2
**Contribution:** 3
**Rating:** 6
**Confidence:** 3

**Summary:**

This paper presents a retrieval task to illustrate the source of the reasoning power of hierarchical transformer models. With the retrieval task, the author takes various structures to verify the connection between successful learning and the presence of an implicit curriculum. In my opinion, this work is helpful in understanding the ability of large language models. As descributed in this paper,  a transformer cannot solve the retrieval problem with a specific difficulty unless it has the minimum number of necessary layers.

**Strengths:**

This is a good joy for understanding the ability of LLM, especially the emergent ability of models.


1.  a novel idea to study the ability of the reasoning ability of LLM.

2. The finding is exciting and fits with human intuition.

3. The visualization of attention can give describution on how LLM takes retrieval reasoning tasks.

**Weaknesses:**

1. Can you provide a more complex example?

2. In my opinion, I hope I can see a general framework that can unify more tasks with your retrieval task.

3.  There are a lot of chapters in the article, and I can't quite understand the relationship between different chapters.

**Questions:**

Can you describe the definition of the implicit curriculum in detail? In your experiments, I can see the connection between the number of layers and the model's performance, as shown in Figure 4.

The emergent ability of LLM is more depend on depth or width?  This paper mainly analyzes the effect of model depth on emergence ability.

---

> ### Author Response · Authors · 2024-11-16
>
> Thank you very much for your review! We provide answers to your questions below.
>
> > Can you describe the definition of the implicit curriculum in detail?
>
> The implicit curriculum (IC) formulation is described on lines 235-237. In the IC formulation, the target vectors have DK dimensions and contain all the tokens forming the retrieval chain concatenated (except the query token x0). For example, for the input sequence "CD…BC…AB…A", the IC formulation would predict the concatenation of [B, C, D].
>
>
> > The emergent ability of LLM is more dependent on depth or width?
>
> We did some experiments with varying widths, but we observed that width does not play a large role in the retrieval problem. However, we believe that this depends on the problem. There might be other problems where width is just as important. For example, we can think of a problem where the model must store and work with very high dimensional data.
>
>
> > Can you provide a more complex example?
>
> Sure, what kind of example would you like? The Kingdoms formulation provides maybe the most realistic complex example (5 retrieval steps): “Alice lives in Novaria. Novarians believe in harmonianism. Harmonianists eat lamb. Lamb contains Zephyrium. Zephyrium causes Chronogy. Who has Chronogy?”
>
> Of course, the names and places are not real, but we could easily imagine how a similar chain of information could occur in a real-world text corpus.
>
>
> > In my opinion, I hope I can see a general framework that can unify more tasks with your retrieval task.
>
> Could you be more specific on which tasks you believe could be unified with our task?
>
> We share your interest in unification and generality. The retrieval task is very general compared to the original induction task. The conditional retrieval variant is particularly powerful.

---

> ### Author Response · Authors · 2024-11-30
>
> As the discussion period is soon drawing to a close, we would greatly appreciate if you could provide a response to our rebuttal and updated paper. Thank you!

---

### Official Review · Reviewer_MJk6 · 2024-11-03

**Soundness:** 3
**Presentation:** 2
**Contribution:** 3
**Rating:** 8
**Confidence:** 4

**Summary:**

The paper proposes a novel retrival problem to conduct an empirical but theoretically grounded anaylsis of mechanisms leading to reasoning in transformers.
Expanding on the induction problem the authors generalize the task to an arbitrary number of inductions steps and introcude a conditional variant.
The authors demonstrate that current SOTA LLMs struggle with the retrival problem and their performance approaches random guessing with increased problem complexity.
Experiments on a minimal problem formulation indicate the importance of implicit curricula and highlight a correlation between the difficulty of the task and the number of layers.
Through manual inspection and subsequent experiments the authors identify learned circuits in a subset of attention heads which emerge one head at a time during training.

**Strengths:**

- The paper makes a strong contribution to the field of mechanistic interpretability and furthering our understanding of the transformer architecture and training behavior. Especially the finding on the sequential emergence of attention heads for reasoning circuits is valuable.
- The **problem statement** at hand is presented nicely, and the paper follows a logical progression building up to the final insights.
- The **structure and flow of the experiments** are sensible, starting with higher-level analysis and ending in more focused experiments.
- Furthermore, the experiments were conducted rigorously.

**Weaknesses:**

**Readability:**
- Section 5 "THEORETICAL ANALYSIS OF INFORMATION FLOW" is quite **hard to follow** and requires some time to understand, especially with limited prior knowledge. Concrete, examples to what "E", "F", ... and so forth might mean in the training and evaluation context could help the reader to grasp the theoretical analysis quicker. Augmenting this section with examples, e.g. from  Sec. 4, would make it more accessible to a larger audience.

**Missing clarity:**
- Large portions of the paper's analysis are **based on a strong assumption**. Key aspects are *Assumption 1* along with simplification of the transformer architecture. The authors should provide further justification for the reasonability of these assumptions and why they would generalize to LLMs. For example, an elaboration on the "shared token information" assumption in real-world scenarios would benefit the paper.
- The **real-world example from the discussion section could be used earlier** to better motivate the theoretical analysis and problem statement.

**Generalization:**
- Similarly, the paper would benefit from an analysis on a real-world question-answer dataset, for example.

**Method:**
- Lastly, the **method of _manually_ reverse engineering** the circuits is only mentioned but **not described in detail**. This severely affects the potential replicability of the paper and makes independent assessments of the methodology challenging.


**Post-rebuttal edit**
The rebuttal addresses the main weaknesses of the paper. I adjusted my score accordingly.

**Questions:**

1. Can the authors elaborate on why Assumption 1 should hold for LLMs, as well as how their
2. From section 4 and Figure 2, I take that all variations of the problem formulations (F1 to F4) include 4 choices, given the random guessing probability of 25%. However,  this does not seem to be the case for F5 "Relatives". Is there a specific reason for that?
3. Do the authors have an intuition of how their results might be generalized to real-world data? It would be valuable to asses if reasoning circuits follow similar emergence patterns during training. However, I assume that the manual identification of circuits is not scalable.
4. What methods were used to "reverse engineer" the attention head circuits?

---

> ### Author Response · Authors · 2024-11-16
>
> Thank you for the insightful review and suggestions! We provide answers to your questions below.
>
> > Can the authors elaborate on why Assumption 1 should hold for LLMs.
>
> Assumption 1 follows directly from the architecture of the transformer. During self-attention, a position A can attend to another position B only if A's query vector aligns with B's key vector. This key-query alignment is only possible if the residual streams of A and B already share some information. More precisely, the shared information is located in the row spaces of the Query and Key matrices of A and B, respectively.
>
> In LLMs, this "shared information" can be a token embedding, a positional encoding, or any nonlinear combination or transformation thereof. In the context of the retrieval problem, we only consider pure token embeddings and positional econdings since they cannot be meaningfully combined or transformed to obtain "new shared information".
>
> Thank you for highlighting this lack of clarity! We will make sure to add a discussion and justification of our assumptions.
>
> > From section 4 and Figure 2, I take that all variations of the problem formulations (F1 to F4) include 4 choices, given the random guessing probability of 25%. However, this does not seem to be the case for F5 "Relatives". Is there a specific reason for that?
>
> Yes, for the "Relatives" formulation we decided to go with a prompt with 16 different choices. This is due to the nature of the task (4 chains x 4 relatives per chain = 16). Feel free to take a look at Appendix A for the complete prompts.
>
> > Do the authors have an intuition of how their results might be generalized to real-world data? It would be valuable to assess if reasoning circuits follow similar emergence patterns during training. However, I assume that the manual identification of circuits is not scalable.
>
> We believe that it should be possible to find retrieval heads in real-world LLMs using a combination of manual and semi-automated interventions, similar to the work of Wang et al (2022). Once a retrieval head has been identified, its emergence during training could be studied using our method from Section 9, assuming that training checkpoints are available.
>
> > What methods were used to "reverse engineer" the attention head circuits?
>
> The precise method used is the following:
>
> 1. First, attention maps were plotted for each head for several prompts.
> 2. Then, we generated a list of hypothesized mechanisms for each head based on the attention maps. This was quite straightforward since most attention maps revealed a clear pattern. You can see the attention maps in Appendix B.
> 3. Then, we found the correct mechanism for each head using ablations.
> 4. Then, we validated the complete mechanism by performing all ablations together.
> 5. Finally, we validated that our mechanism is not excessive by attempting to perform further ablations.
>
> We will add the full method to the paper. In principle, there is no reason why our method could not be automated. In fact, it is very reminiscent of automated causal interventions, which could have also been used successfully. But since our models were rather small, we decided that using automated methods would not serve us.
>
> > Similarly, the paper would benefit from an analysis on a real-world question-answer dataset, for example
>
> Could you please be more specific on what kind of analysis you would like to see on a question-answer dataset that would be relevant within the context of our work?

---

> > ### Comment · Reviewer_MJk6 · 2024-11-25
> > **Rebuttal Adresses Key Weaknesses**
> >
> > Thank you for the detailed response.
> >
> > Taking into consideration the other reviews, responses, and revised version of the paper, I recommend acceptance to ICLR.
> >
> > I have adjusted my score accordingly.

---

### Author Response · Authors · 2024-11-22

We would like to thank all reviewers for their insightful feedback!

We have updated our paper to address the two most important criticisms:

1. We have **weakened assumption 1** and improved its justification
2. We have added a detailed description of the **reverse-engineering procedure**

We changed assumption 1 from stating that attention requires a "shared token embedding or positional encoding" to stating that it requires a "shared piece of information" (any information). This assumption is motivated by the fact that a position can only attend to another position if their key and query vectors align. Constructing aligned key-query pairs is only possible if the two positions already share some information. More precisely, the shared information must be located in the row spaces of the query and key matrices of the attending and attended positions, respectively.

To maintain the soundness of the argument, we have introduced the old assumption as a lemma dervied from the new assuption. Its proof is based on the fact that the token values and pair positions are assigned randomly and independently.

Regarding the reverse-engineering procedure, we have added a comprehensive description in Appendix D. We have decided to place it in the appendix (rather than the main paper) since the procedure is not particularly innovative or interesting. In fact, we believe that a great number of different procedures could have been used to arrive at the same circuits. Nonetheless, including the procedure is important for reproducibility.

We would like to thank the reviewers for highlighting these issues and we hope that you will appreciate these improvements!

If the paper is accepted, we will fix the remaining issues in the presentation and clarity of the paper, as highlighter by the reviewers. We are also happy to include additional experiments. Any suggestions in this regard are highly appreciated!

---

### Meta-Review · Area_Chair_xW4s · 2024-12-22

**Metareview:**

This paper analyzes a retrieval problem, that involves extracting information from multiple positions within an input sequence. The authors show that solving this problem necessitates a specific depth of Transformer layers. The study investigates how large language models and transformers approach retrieval tasks, introducing two variants: the retrieval problem and the conditional retrieval problem. It reveals that large language models can tackle retrieval tasks without fine-tuning by utilizing sophisticated mechanisms involving multiple attention heads, which naturally emerge in a specific order when trained using an implicit curriculum. There is also theoretical analysis for this.
The paper tackles an interesting topic in the LLM interpretability domain and demonstrates results not only for toy LLMs, but also sota LLMs like Gemini etc. The reviewers particularly find the experimental studies, the clear results and the interesting findings of the role of an implicit learning curriculum as strong points of the paper. As weaknesses, there were some initial questions about insufficient experimental details and lack of too strong assumptions, the hard-to-follow theoretical section 5 and lack of results on real-world Q&A datasets.  Given the clear idea and execution and the additional clarifications provided in the rebuttal phase, the AC recommends acceptance.

**Additional Comments On Reviewer Discussion:**

The authors addressed most points from the reviewers, such as experimental details, information on assumptions and theorems (yct3) and further clarifications (cnTJ, MJk6) and lack of details on the reverse engineering procedure (MJk6). Some points about lack of real-world examples/datasets remains, similarly the presentation of the paper can further be improved. But overall, the AC finds that this paper meets the bar of acceptance for ICLR.

---

### Decision · Program_Chairs · 2025-01-22

Accept (Poster)